# Experimental Study on the Movement and Evolution of Overburden Strata Under Reamer-Pillar Coal Mining Based on Distributed Optical Fiber Monitoring

**Chunde Piao [1,\*], Shaogang Lei [2], Jiakun Yang [1,3] and Lihong Sang [4]**

1   School of Resources and Geosciences, China University of Mining and Technology, Xuzhou 221116, China; ts15010095a3@cumt.edu.cn
2   School of Environment Science and Spatial Informatics, China University of Mining and Technology, Xuzhou 221116, China; lsgang@126.com
3   China Construction First Building (Group) Corporation Limited, Beijing 100071, China
4   School of Land Science and Technology, China University of Geosciences, Beijing 100083, China; sanglihong2009@163.com
\*   Correspondence: piaocd@cumt.edu.cn; Tel.: +86-0516-83591000

**Abstract:** Focusing on the deterioration of the surface ecological environment caused by large-scale exploitation of the Jurassic coal field in northern Shaanxi, the three-dimensional similar material test model is made to simulate the extraction of shallow coal seam. Using Brillouin optical time-domain analysis (BOTDA) optical fiber distributed sensing technology, this paper studied the strain distribution rule and movement characteristics of strata under reamer-pillar mining, analyzed the stability of the remaining coal pillars in the mining area, and obtained the strain contour graph of strata through calculations. The research result shows that the deformation of coal pillars in a safety-critical state under reamer-pillar mining experiences three stages. The stratum deformation is distributed in the shape of a pyramid with the mining area as the center. On the basis of the strain distribution of strata, the settlement curve and subsidence curve of strata deformation are determined to obtain the rupture angle and angle of draw. After being compared with the measured data, the angle values are almost the same.

**Keywords:** BOTDA; optical fiber sensing; reamer-pillar mining; shallow coal seam; stratum movement

## 1. Introduction

The Jurassic coal field in Yushenfu of northern Shaanxi is the largest proven coal field in China, whose amount of coal resources accounts for about 14% of the total retained coal resources in China [1]. As the mining area is located in the arid area of northwest China and at the border of the Mu Us Desert and Loess Plateau, its water resources are scarce, vegetation is sparse, and the surface ecological environment is fragile. Reamer-pillar coal mining refers to the process in which coal pillars (columns) of a certain width are created to support the roof when mined material is extracted a certain distance across the longwall coal face, thus preventing the coal mine roof from being expanded, effectively controlling the surface subsidence and implementing water-preserved mining.

The stability of coal pillars in the reamer-pillar mining method controls the deformation of overburden rock. At present, through theoretical analysis and numerical simulation, the rationality of setting coal pillar parameters in reamer-pillar mining is discussed, but the selection of rock mass parameters and the applicability of theoretical formula need to be further improved [2,3]. Overburden

deformation and failure induced by mining is a dynamic phenomenon of stress, strain, and energy conversion. By monitoring overburden deformation, the parameter characteristics of rock–soil body change can be effectively identified [4]. An install inclinometer [5], a groundwater level manometer [6], an optical fiber sensor [7], and a borehole TV tester [8] are installed in overburden strata of the coal mining face to grasp the movement characteristics of overburden strata in time. Important parameters such as strain and displacement for the mining subsidence prediction are obtained through the coring in surface boreholes [9] and the statistical theory of big data [10]. The geomechanic model test, as an effective means of rock engineering research, has the advantages of simulating a geological environment, stratum subsidence process, and time effect. Therefore, it has been applied in the study of the deformation mechanism of overburden mines [11,12]. Digital photography and ground laser scanning technology [13], BOTDR (Brilliouin optical time domain reflectometer) technology [14], a fiber Bragg grating sensor [15], and a stress sensor [16] are employed to continuously monitor the overburden deformation in the coal mining model test, analyze the development characteristics of mining fissures, and determine the reasonable sizes and stability of coal support pillars. In the above tests, similar materials are laid by models, which can only analyze the plane deformation problem, and it is difficult to grasp the overall deformation characteristics of overburden rock in the mining area. There is no reliable and simple method to determine subsidence parameters based on field monitoring data.

In this paper, with a self-made 3D model testing box and the overburden strata and soil layers of coal face 52303 in Daliuta mine as the research object, BOTDA (Brillouin optical time domain analysis) optical fiber distributed sensing technology is used to study the strain distribution and movement mechanism of overburden strata under reamer-pillar mining and to explore the way to determine the settlement parameters of strata during coal mining. It can provide useful references for safe exploitation of coal resources and proper protection of land resources in the western mining area of China.

## 2. Similar Material Model Test Scheme

### 2.1. Model Scheme Design

The study area is located in Daliuta Coal Mine, Daliuta Town, Shenmu County, Shaanxi Province, along the Wulanmulun River. Its geographic coordinates are (N39.1~N39.4, E111.2 ~E110.5). The overburden strata of coal face 5230 is used as the research object in this test. Through the drilling test, the buried depth of the coal seam is 235 m and its thickness is 6.9 m. The stratum structure of the face consists of three parts: the Quaternary unconsolidated layer (thickness: 30 m), fine sandstone (thickness: 187 m), and siltstone (thickness: 18 m) from the top to the bottom, respectively. The main section of the 52303 working face is shown in Figure 1.

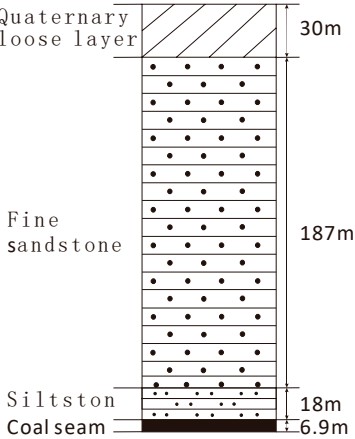

**Figure 1.** The section figure of the 52303 work facing.

According to the prototype of coal face 52303 and experimental site conditions, the model test device is designed to be 3.5 m long, 3.1 m wide, and 0.77 m high. According to the similarity between the test model and the site prototype, the geometric similarity ratio in this test is 1:300, the density similarity ratio is 1.23, the stress similarity ratio is 369, and the strain similarity ratio is 1. Assuming that its deformation meets the brittleness requirements of materials, the deformation and failure characteristics of overburden rock are analyzed according to maximum normal stress theory. The mechanical parameters of the similar strata are shown in Table 1.

**Table 1.** The mechanical parameters of similar rock material.

| Stratum | Prototype | | | | | Model | | | | |
|---|---|---|---|---|---|---|---|---|---|---|
| | Thickness (m) | Density (g/cm$^3$) | Compressive Strength (MPa) | Tensile Strength (MPa) | Elastic Modulus (GPa) | Thickness (m) | Density (g/cm$^3$) | Compressive Strength (MPa) | Tensile Strength (MPa) | Elastic Modulus (GPa) |
| Fine sand | 187 | 2.64 | 30.60 | 4.28 | 42 | 0.62 | 1.50 | 0.082 | 0.012 | 114 |
| Silty sand | 18 | 2.58 | 29.20 | 4.06 | 40 | 0.06 | 1.47 | 0.079 | 0.011 | 108 |

The average dip angle of overlying strata in the 52303 working face of Daliuta Coal Mine is 2°, and no fault passes through the study area. The physical and mechanical properties of silty and fine sand in the overburden strata are close to each other. In order to facilitate the laying of the model, this test would regard fine sand and silty sand as the same rock layer during the testing process, and the laying inclination of each rock layer in the model is set as 0°. Considering the joint fissures in the rock mass and in order to ensure the homogeneity of the model laying, layered paving is adopted. After each layer (thickness: 0.03 m) is paved, the mica is spread on it so as to ensure that the stratum stratification is clear. The aggregate is sand and mica powder, the cement is gypsum and calcium carbonate, and the borax serves as the retarder. Under the prerequisite that the similarity ratio of the compressive strength of each rock layer is fully considered, the ratio of the sand/mica powder/gypsum/calcium carbonate is 0.80:0.18:0.014:0.006. After the ingredients are mixed based on the ratio, 10% of the total amount of water is added for stirring. The similar material composition and material usage of each rock layer are shown in Table 2.

**Table 2.** The ratio and dosage of similar material.

| Strata | Total Thickness (m) | Layers | Thickness (m) | The Amount of Material per Layer (kg) | | | | | |
|---|---|---|---|---|---|---|---|---|---|
| | | | | Sand | Mica Powder | Gypsum | Calcium Carbonate | Water | Borax |
| Fine sandstone | 0.62 | 21 | 0.03 | 513.36 | 115.51 | 8.99 | 3.85 | 39.02 | 0.64 |
| Siltstone | 0.06 | 2 | 0.03 | 513.36 | 115.51 | 8.99 | 3.85 | 39.02 | 0.64 |

## 2.2. Distributed Optical Fiber Sensing Technology and Model Testing Method

### 2.2.1. BOTDA Distributed Optical Fiber Sensing Technology

The detection principle of BOTDA is that step-pumped pulsed light and continuous light are injected into both ends of the fiber. When the frequency difference between the two is equal to Brillouin frequency shift, stimulated Brillouin scattering will occur. When the deformation of a section of the optical fiber occurs or the temperature changes, the Brillouin frequency of the section will drift [7,17]. The monitoring schematic diagram of BOTDA is shown in Figure 2.

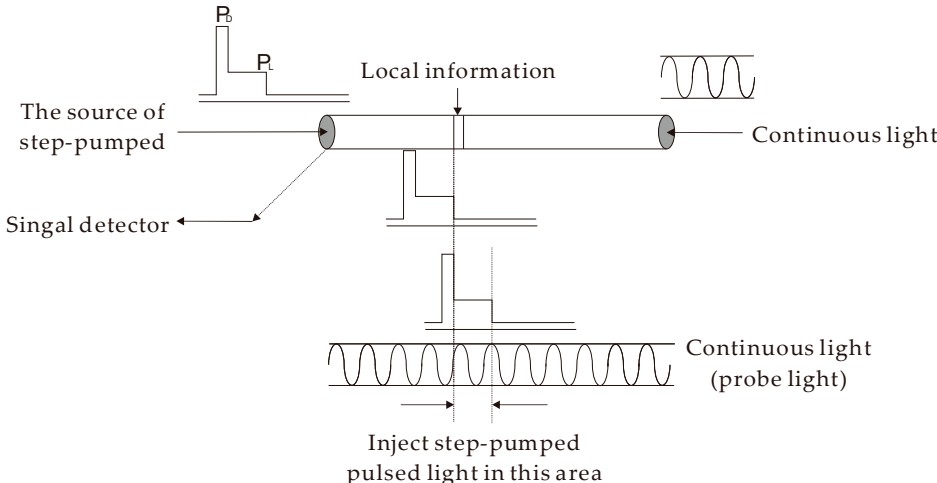

**Figure 2.** Monitoring schematic diagram of Brillouin optical time-domain analysis (BOTDA).

According to the linear relationship between Brillouin frequency shift and strain and temperature, the strain and temperature at each point along the optical fiber can be determined. The relationship between Brillouin frequency shift and strain and temperature is shown in Equation (1).

$$v_B(\varepsilon, T) = v_B(0) + \frac{dv_B(\varepsilon)}{d\varepsilon} \cdot \varepsilon + \frac{dv_B(T)}{dT} \cdot (T - T_0) \tag{1}$$

In the formula, $v_B(\varepsilon, T)$ is the drift of Brillouin frequency with strain or temperature; $v_B(0)$ is the drift of Brillouin frequency without strain or temperature; $\varepsilon$ is the axial strain of sensing fiber; $(T - T_0)$ is the change of external temperature; and $dv_B(\varepsilon)/d\varepsilon$ and $dv_B(T)/dT$ are the influence coefficients of strain and temperature, respectively.

In this paper, the NBX(NEUBRESCOPE™)-6000 PPP-BOTDA(Pulse-PrePump Brillouin Optical Time Domain Analysis) analysis developed by Kobe of Japan is used to monitor the overburden deformation, and the main technical performance indicators of the analysis in the monitoring process are shown in Table 3.

**Table 3.** Main technical performance indicators of NBX-6000 in test model monitoring.

| Measuring Range (km) | Spatial Sampling Interval (m) | Frequency Sampling Interval (MHz) | Average Times | Spatial Positioning Accuracy (m) | Strain Measurement Range | Pulse Width (ns) | Spatial Resolution (cm) | Strain Measurement Accuracy (µε) |
|---|---|---|---|---|---|---|---|---|
| 1 | 0.05 | 5 | $2^{13}$ | $\pm(2.0 \times 10 - 5 \times$ measuring range (m) + 0.2 m + 2 × distance sampling interval (m)) | −3%~4% | 1 | 10 | ±25 |

### 2.2.2. Excavation and Test Method of Test Model

The strip mining is simulated by extracting steel bars in the center of the model box. The length, width, and height of each bar are 1100 mm, 60 mm, and 23 mm, respectively. The width of each coal pillar between each steel bar is 22 mm and it is filled with similar materials to simulate reamer-pillars. Each excavation takes 30 minutes, and the test is carried out after the stratum is stable for 30 minutes. The test is divided into 12 steps of strip mining, and the total mining distance is 984 mm.

In order to understand the characteristics of overburden deformation caused by coal seam mining, according to the "three zones" discriminant theory, three monitoring layers (I, II, and III) are set up along the strike of the working face at 5 mm, 205 mm, and 405 mm away from the top of the coal seam, respectively [18]. According to the disturbance degree caused by coal seam mining, two sensing optical fibers are laid along the center of goaf, the edge of goaf, and non-goaf in each monitoring layer. Before

the test, the sensing optical fiber at the adjacent position of each layer is fused from the beginning to end. Then, the sensing optical fiber of each layer is connected in series from top to bottom to form one optical fiber. BOTDA is used to monitor the horizontal strain caused by mining overburden subsidence. The size of the model box, the arrangement and connection mode of the optical fiber sensor, and the steel bar setting mode are shown in Figure 3.

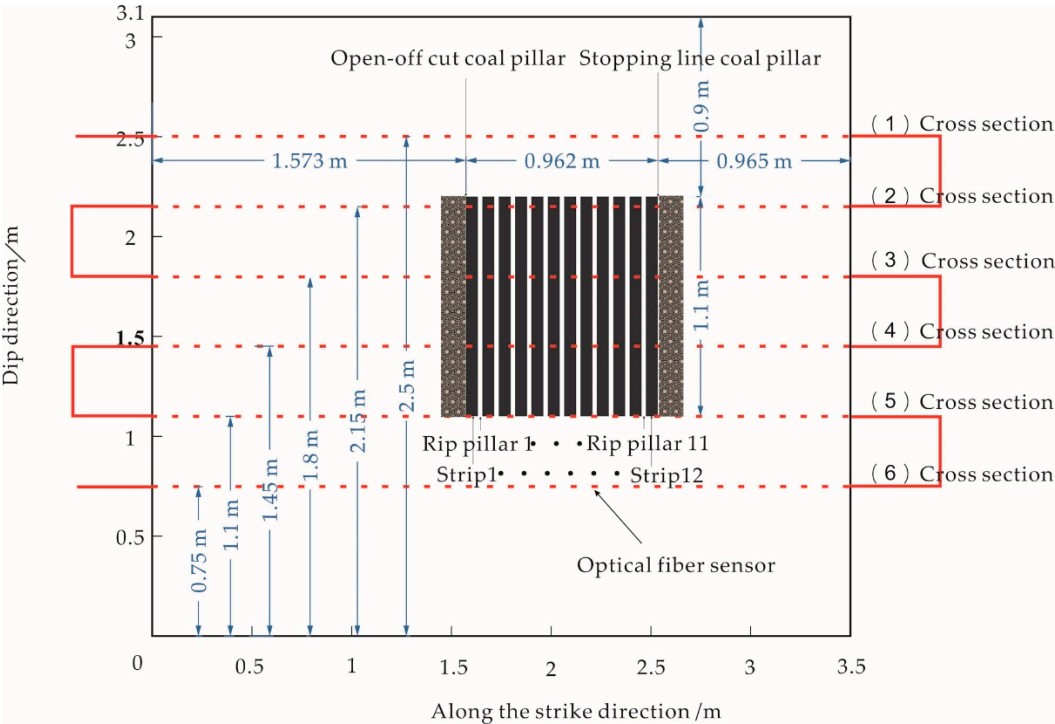

**Figure 3.** Model box size and layout optical fiber sensor.

The model test is conducted in a closed laboratory, and the temperature barely changes during the test. A loose sensing optical fiber is fixed in the metal bellows as a temperature compensation sensor, which is installed in the internal model during the model laying [19]. During data processing, the strain caused by temperature change is subtracted from the strain of overburden caused by coal seam mining. Therefore, the impact of temperature change on test data is eliminated.

## 3. Experimental Result and Analysis

### 3.1. Strain Analysis of Reamer-Pillars in the Exploitation Area

The strain distribution of overburden strata during the process of reamer-pillar mining can be obtained through optical fiber sensors laid in the model test. In this article, because of the article length and convenience of analysis, the strain distribution of the 1st, 3rd, 5th, 7th, 9th, 11th, and 12th pillars in layers I and II of the third section during excavation is used to analyze the deformation features of overburden strata. Figures 4 and 5 show the strain distribution of each pillar.

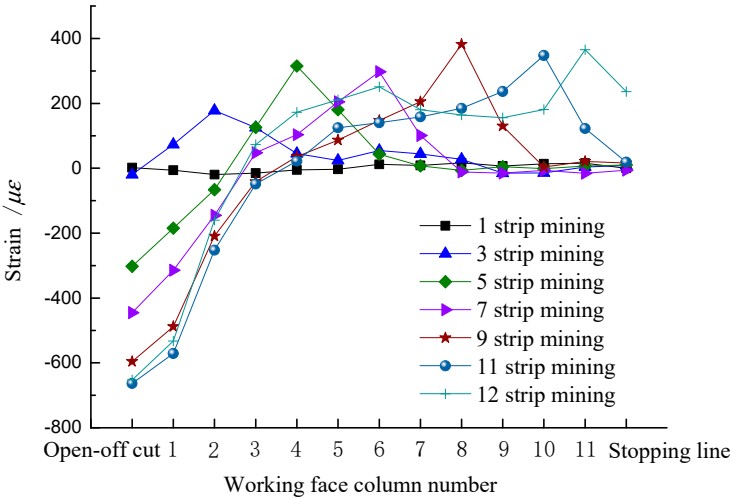

**Figure 4.** Strain change and distribution map of each pillar in layer I along with excavation of the coal face.

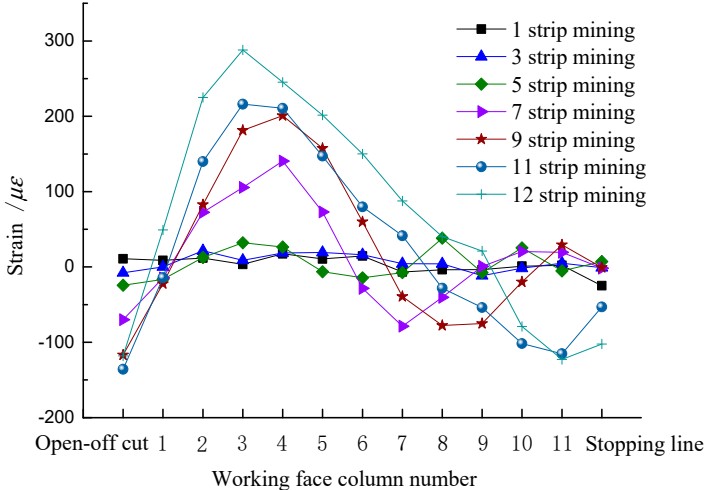

**Figure 5.** Strain change and distribution map of each pillar in layer II along with excavation of the coal face.

As shown in Figure 4, when the first strip is being mined, the deformation of each pillar is small and the strain approaches zero, which means that the overburden strata do not fall apart easily because of the support of the coal pillars on both sides of the goaf. When the third strip is being mined, the deformation of reamer pillar 2 increases, which means that the roof of the goaf where the second and third strips are located collapses. However, the roof does not break down completely because of the support effect of pillar 1 and 3, and the disturbance range of strata is limited. According to the physical and mechanical parameters of Daliuta coal field and the calculation formula of pillar stability, it could be obtained that the pillar strength is 6.714 MPa, the vertical stress that a pillar bears is 6.435 MPa, and the calculated safety factor is 1.04, which means that the deformation of coal pillars is in critical state [20,21]. The peak stress of the coal pillar exceeds the tensile strength of the material when the fifth strip is being mined. The peak tensile strain of the reamer pillar behind the working face occurs because of the successive collapse of the overlying strata, and the maximum strain is 350–400 μ$\varepsilon$. With the gradual loss of the carrying capacity of coal pillars, "domino" instability collapse occurs—the strata on both sides of the goaf move to the center of goaf, causing compressive strain of coal pillars and reamer pillar 3 on the left side of the open-off cut. The compressive strain gradually increases with the increase in the mined amount. However, the strain growth gradually reduces and tends to be stable.

For the coal pillars in the unmined area in the front of the coal face, with the support of the entire stratum structure, the roof deformation range is gradually reduced from the internal of the goaf to the external.

It can be seen from Figure 5 that for layer II, when the fifth strip is being mined, the strain change tends to be 0, indicating that the height of the caving zone caused by mining has not reached the length of layer II. The peak stress of the coal pillar exceeds the tensile strength of the material when the seventh strip is being mined. Affected by the deformation, fracture, and caving of the main roof, the peak tension strain of the reamer pillar appears in the middle of the mining area, and the surrounding strata move horizontally to the goaf. At the same time, the tensile strain occurs at the center of the mining area as a result of the settlement of strata, and the compression stress zone is formed at the open-off cut and the upper stratum of the reamer pillar in the front of the coal face. With the increase in the number of mined strips, the range and the strain value of the tensile stress zone increase gradually and more slowly, and the compressive stress zone gradually moves toward the open-off cut and stopping line, but the value growth reduces.

### 3.2. Strain Distribution Characteristics of Overburden

#### 3.2.1. Strain Distribution Characteristics of the Horizontal Profile

Through the six optical fiber sensors laid in layer I and II, the contours of the strain at each layer are obtained. In this paper, because of the length limitation, the deformation of the overburden strata is analyzed by contours of the strain when mining the 2nd, 5th, 8th, and 11th strips. Figures 6 and 7 are the strain contour maps of layer I and II, respectively under reamer-pillar mining. Among them, the red box marked is the accumulated range of strips mined along the coal face.

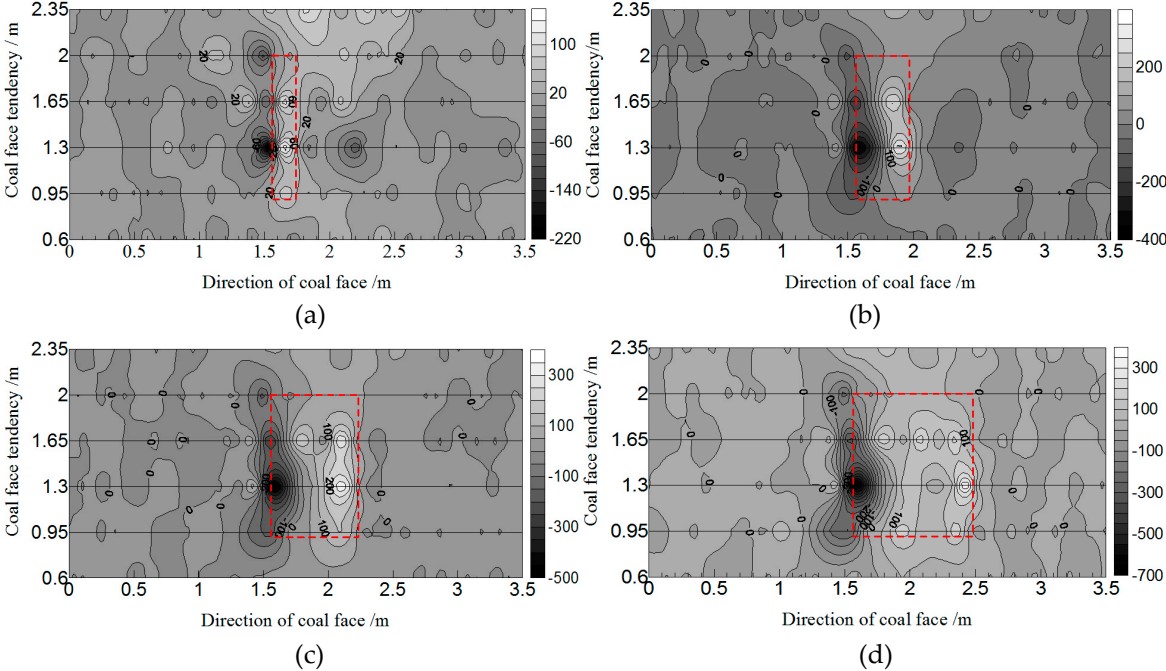

**Figure 6.** Stratum strain contour map of layer I: (**a**) the 2nd strip mined; (**b**) the 5th strip mined; (**c**) the 8th strip mined; and (**d**) the 11th strip mined (unit: $10^{-6}$).

The deformation characteristics of overburden strata in layer I are along the strike of the working face. When mining the second strip in Figure 6a, the bending deformation and tensile strain of the strata in the working face range are caused by coal mining, the maximum value is 120 $\mu\varepsilon$, and the overlying strata are in the critical state of failure. The compressive stress concentrates in the coal pillar

area behind the open-off cut. Combined with the strain value of the coal pillar behind the open-off cut, it can be induced that the disturbance range of the strata is 69 m, and the maximum strain value is $-220$ µε; therefore, the strata have not been destroyed. When the fifth strip is mined in Figure 6b, two stress zones are formed in the disturbed area of overburden strata. The first is the compressive stress zone between 45 m of coal pillar behind the open-off cut and reamer pillar 3, with the maximum strain of $-400$ µε; the second is the tension stress zone between reamer pillar 3 and 5, with the maximum strain of 300 µε. The damage range of rock strata is up to 60 m. From Figure 6c,d, it can be seen that with more and more strips being mined, the compressive strain area is between 45 m of coal pillar behind the open-off cut and reamer pillar 3. The compressive strain zone does not expand, but the compressive strain value increases gradually. The tension strain zone is located between strip 4 and the coal pillar behind the working face. Its damage area is further expanded. Along the working face tendency, the strain of the goaf and the overburden rock in the center of the reamer-pillar is the largest because of the disturbance of coal seam mining, and gradually decreases to zero towards the edge of the goaf. With the increase of the reamer-pillar mining face, the stress concentration phenomenon at the edge of the goaf lags behind the center of the goaf, and the strain value of the overburden rock distributes symmetrically along the center of the mining face.

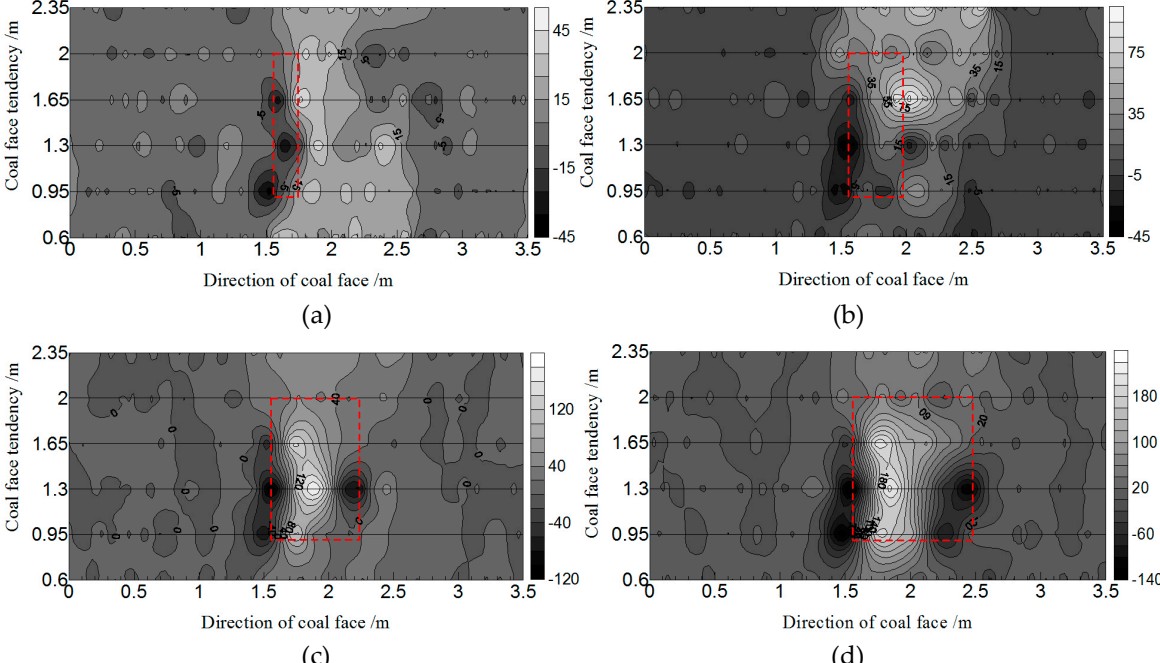

**Figure 7.** Strain contour map of layer II: (**a**) the 2nd strip mined; (**b**) the 5th strip mined; (**c**) the 8th strip mined; and (**d**) the 11th strip mined (unit: $10^{-6}$).

The deformation characteristics of the overburden strata of layer II are as follows: Along the strike of the working face, when the second strip in Figure 7a is being mined, the disturbance effect of coal mining on the rock mass of layer II is small. From Figure 7b–d, it can be seen that the compressive stress concentration occurs in the overlying strata at the center of the cut hole and stopping line of the coal seam from the fifth strip of coal seam mining. With the increase in the number of mining strips, its magnitude increases gradually, and the compressive stress area is between 40 m behind the cut hole and cutter pillar 1. With the mining of strips, the bending settlement of overlying strata results in a tension stress zone, but no strata damage occurs. Along the inclination of the working face, with more and more strips being mined, the stress concentrates in the goaf and at the center of reamer pillars gradually. Similar to the stress of overburden strata in layer I, the strain of overburden strata in the

central area is at a maximum, then gradually decreases towards the edge of goaf, but the strain value decreases significantly compared with layer I.

### 3.2.2. Strain Distribution Characteristics of Vertical Profile

Along the inclination of the working face, through the optical fiber sensors at layers I, II, and III on the third section, the strain distribution after all 12 strips are mined is obtained, as shown in Figure 8.

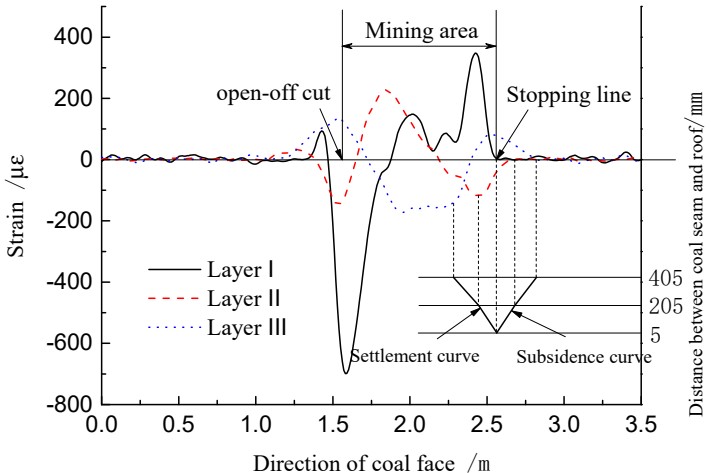

**Figure 8.** Vertical strain distribution of overburden strata.

It can be seen from Figure 8 that the area 15 m away from the roof at layer I belongs to the caving zone and its deformation is divided into two parts. The first part, surrounding the strata between pillars behind the open-off cut and reamer pillar 3, moves horizontally towards the goaf, causing the compressive strain. The second part is in the area between the fourth strip and the stopping line. Because of deformation of strata in the goaf and the support of reamer pillars, the strain value wavily changes. The area 61.5 m away from the roof at layer II belongs to the fracture zone. The tensile strain is generated as a result of the stratum settlement in the middle of the mining area. The compressive strain is generated in the middle of mining area as a result of strata subsidence, and compressive strain occurs from the horizontal as a result of the horizontal compression of strata from the open-off cut to the stopping line. The strain curve is in the shape of a "W". For layer III, the rock disturbance in the area 121.5 m away from the roof is small; therefore, the compressive strain zone in the middle of the mining area and tensile stress zone in the boundary are formed. The strain distribution is in the shape of a "U".

The settlement curve and subsidence curve of overburden strata are obtained through the optical fiber sensors and strain distribution graphs. According to the strain distribution characteristics of the strata of various heights from the stopping line to the roof, the rupture angle and angle of draw in this test are 56.7° and 57.1°, respectively [22].

## 4. Deformation Characteristics of Overburden Strata

(1) Based on the Willson formula of progressive damage, the width of the pillar yield zone in this model is 6.72 m, which is bigger than the width of the reserved pillar (6.6 m) [18,21]. At the beginning of reamer-pillar mining, similar to longwall coal mining, the deformation of the strata is relatively small because of the "hinged arch" effect between the support of the reamer pillars and the overburden strata [23]. With more and more strips being mined, the settlement amount and deformation range of the overburden strata caused by deformation and failure of reamer pillars expand gradually so that the strata in the caving zone move toward the goaf. As the distance between the working face and

open-off cut increases, the rock burst deformation and coal pillar compression settlement caused by full mining in the goaf alternately occur in the mining area and the deformation curve is distributed in a wavy shape. The optical fiber sensor at layer II is located at 0.3 H (H is the buried depth of the coal seam) and is at the critical point of rock movement [21]. With the gradual compaction of the caving zone, the movement of rock strata in the fissure zone is similar to that of long-walled mining. At the center of the mining area, the movement is mainly about settlement. At the boundary of the mining area, the deformation pattern is mainly horizontal movement and settlement.

(2) Compared with longwall mining, strain distribution of overburden strata caused by reamer-pillar mining has the following characteristics. Along the inclination of the working face, the deformation of strata in the caving zone can be divided into three stress zones: the rock pressure bearing zone in the early stage of mining, the compressive stress zone of the coal pillar near the open-off cut due to the collapse of the rock, and the wave-shaped tensile stress zone caused by deformation of the reamer pillar. The deformation of the rock within the fracture zone is divided into two stress zones: the tensile stress zone caused by stratum settlement in the center of the goaf and the compressive stress zone caused by the horizontal movement from the boundary to the center of the goaf. Along the inclination of the coal face, the deformation of overburden strata is divided into three stress zones, that is, the stress concentration area between the goaf and reamer pillars, the stress reduction area from the middle of the mining area to the boundary, and the stress area of the original rock outside the mining boundary. They are distributed in an "isosceles triangle" shape.

(3) The model test is mainly made of compacted gravel materials [13,15,16]. Because of the inhomogeneity of the similar material and the degree of compaction, there is an inconsistency between the material properties of the same rock layer. From this experiment, we can see that the deformation of the rock caused by mining along the working face has certain fluctuations, which should be further improved in the configuration and model making of similar materials in future research.

## 5. Conclusions

To sum up, the following conclusions are obtained in this paper.

(1) The strain distribution of overlying strata under mining is obtained according to the sensor optical fibers laid at different heights. The subsidence curve and subsidence curve of overlying strata deformation under mining conditions are determined. The overlying strata rupture angle and the influence boundary angle caused by coal seam mining are obtained. The derivation method of mining subsidence parameters based on distributed optical fiber sensing technology will provide the necessary calculation parameters for mining subsidence prediction.

(2) The strain distribution of overlying strata based on distributed optical fiber sensing technology shows that when the coal pillars are in critical safety state, the deformation of coal pillars goes through three stages, that is, the deformation support stage of coal pillars in the initial mining stage, the horizontal movement stage of coal pillars near the first mining area caused by the collapse of rock strata, and the "domino" vertical deformation and failure stage of coal pillars between the goafs. Therefore, according to the mechanical properties of overburden, the strength and width of the coal pillar, and the strength of strata behind the cut hole are improved, so that the deformation and failure range of overburden can be effectively controlled.

(3) For the overburden deformation under the condition of reamer-pillar mining, the mining area is taken as the center, and the change is a "pyramid" type. Compared with the longwall mining method, the range and intensity of deformation and failure of overburden strata induced by the reamer-pillar mining are significantly reduced, which is conducive to reducing the surface subsidence area of the mining area and protecting the natural environment of the mining area in western China.

**Author Contributions:** Formal analysis, C.P.; Methodology, S.L.; Project administration, L.S.; Writing—original draft, C.P.; Writing—review & editing, J.Y.

**Funding:** This research was funded by supported by the Fundamental Research Funds for the Central Universities (2017XKQY057), and A Project Funded by the Priority Academic Program Development of Jiangsu Higher Education Institutions (2018).

**Conflicts of Interest:** The funders had no role in the design of the study; in the collection, analyses, or interpretation of data; in the writing of the manuscript, and in the decision to publish the results.

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
