# Peer review of "Experimental Study on the Movement and Evolution of Overburden Strata Under Reamer-Pillar Coal Mining Based on Distributed Optical Fiber Monitoring"

_energies, doi:10.3390/en12010077_

Round 1

Reviewer 1 Report

It is good work for readers to preserve ecological environment of coal mining.

1.      It would be suggested that internationally well-proven references would be cited.

2.      Optical sensing arrangement is not clear. Accordingly it is suggested that optical sensing scheme is described by using the schematic of optical implementation separately from Figure 1, considering the paper was made based on optical measurement. Description of BOTDA is missing such as manufacturer, specification of light source and detectors, and calculation of strain from optical signal, etc.

3.      As was mentioned in the paper, rock and soil parameters are limited to formulate the theoretical model for strain-stress analysis. There is no finding that where and how the material properties appeared in Table 1 were used. Obtaining the material properties by using experiment such as optical fiber sensors, it is suggested that the theoretical model such as stress-strain relation, e.g., possibly by using von Mises criteria to judge whether or not fragile of surface related to strain distribution shown in Figures 4 and 5, could be described in the manuscript. Young’s modulus of the rock material may be missed. It would be more reliablehow to calculate strain distribution.

Thanks!

Author Response

Thank you for your valuable suggestions. According to your suggestions, the contents of the paper have been revised and improved, here are response to the changes.

1. Aiming at the stability of coal pillar, monitoring and evolution mechanism of overburden deformation in the process of cutter-pillar mining. This paper adds eight papers published in international journals, such asInternational Journal of Mining Science and Technology》、《International Journal of Mining Science and Technology》、《Energies》、《International Journal of Rock Mechanics & Mining Sciences》、《International Journal of Rock Mechanics & Mining Sciences. And some related problems are summarized and analyzed.

2. (1) BOTDA distributed optical fiber sensing technology monitoring principle, BOTDA monitoring schematic diagram and strain calculation formula based on this technology is added in section 2.2 (1) of the paper.

(2) The instrument manufacturer (Neubrex Company of Japan), instrument model (NBX-6000 Pulse Pre-Pumped Brillouin Optical Time-Domain Analyzer) and test parameter setting (Table 3) is added in section 2.2 (1) . The specifications of light source and detector can be found according to the instrument manufacturer and product model introduced. Our experiment is applied to research based on this technology, so we are more concerned about the setting of test parameters in the use of the instrument.

3. (1) The tensile strength and modulus of elasticity of two sandstones in prototype and model materials are added in Table 1. The tensile strength and modulus of elasticity of sandstone in the prototype material are obtained by experiment. According to the similarity ratio of the similarity principle, the tensile strength and modulus of elasticity used in the model are calculated.

(2) The use of material attributes in Table 1. Firstly, the method of making similar test model is based on the above material attributes and the proportion of various similar materials, i.e. sand, lime and gypsum. Secondly, the stability of overburden is evaluated based on the compressive strength, tensile strength and modulus of elasticity of the material.

(3) The material used in this paper is based on sandstone, assuming that the deformation meets the brittleness requirement of the material. In the experimental part of the paper, according to "Maximum-Normal-Stress Theory", the rock strain values of Fig. 5 and Fig. 6 (original Fig. 4 and Fig. 5) are compared with the tensile strength and compressive strength of the material in Table 1 by combining the elastic modulus of the material. And the deformation and failure of the mining overburden are analyzed.

Reviewer 2 Report

1) in verse 45, it probably should be ... installing stress sensor mechanism on the top .....

2) there are no references to world research in the work, i.e. no one has ever studied this phenomenon before?

3)few of the referenced works in the bibliography

Author Response

Thank you for your valuable suggestions. According to your suggestions, the contents of the paper have been revised and improved, here are response to the changes.

1. Fig. 5 and Fig. 6 (original Fig. 4 and Fig. 5) are strain contours drawn on the basis of the overburden deformation curves monitored by six sensing optical fibers. The red part in the picture shows the mining range of the coal seam in the reamer-pillar mining.

2. (1) In the second paragraph of the introduction of the first part of the paper, the research background has been revised. Aiming at the problems of coal pillar stability, monitoring and evolution mechanism of overburden deformation in the process of reamer-pillar mining, this paper quotes papers published by scholars in this field in international journals such asInternational Journal of Mining Science and Technology》、《International Journal of Mining Science and Technology》、《Energies》、《International Journal of Rock Mechanics & Mining Sciences》、《International Journal of Rock Mechanics & Mining Sciences. And this paper summarized and analyzed the related problems. In this paper, the latest features of mining overburden deformation, overburden deformation monitoring methods and similar simulation are added, so that readers can understand that the research content expressed in this paper is the inheritance and innovation of previous studies.

(2)  At present, the research on the deformation characteristics of the overburden of the reamer-pillar mining method is based on the two-dimensional similar material test model of plane strain problem. It is easy to be affected by size, and it is difficult to fully understand the characteristics of overburden deformation caused by mining. In addition, there are few reports on the determination of mining subsidence parameters based on distributed optical fiber sensing technology.

3. According to your suggestion, the introduction is revised according to the research content of the paper, which supplements the international reports on related issues and nine papers of scholars in this field have been added.

Reviewer 3 Report

The article is aiming at the deterioration of surface ecological environment caused by large-scale exploitation of Jurassic coal field in northern Shaanxi. In order to achieve this aim the three-dimensional similar material test model is made to simulate the extraction of shallow coal seam. 

The methodology is adequate and the results are orginal. However, the introuduction needs wider context and should include the international study on movement and evolution of overburden strata. Additionaly, the conclusions should be presented in the frame of universality and allowto answer how we could use them in environmental protection and in other coal mining rgions.

Author Response

Thank you for your valuable suggestions. According to your suggestions, the contents of the paper have been revised and improved, here are response to the changes.

(1) Aiming at the research content of this paper, it supplements the research results of the authors in this field on the stability of coal pillars, monitoring and evolution mechanism of overburden deformation in the process of reamer-pillar mining. After summarizing the related problems, the existing problems in the current research are analyzed.

(2) The conclusion of the paper has been revised, and the significance of the research results to the relevant specific projects has been increased. On the one hand, the results can improve the strength of the coal pillars and the overlying strata of the cut hole in the process of reamer-pillar mining. By reducing disturbance to overburden deformation, it can protect aquifer water resources and surface ecological environment. On the other hand, the calculation method of mining subsidence parameters based on optical fiber sensing technology can provide a good idea for obtaining parameters directly in the engineering site.

Reviewer 4 Report

Indicate with a map the situation of the samples (Daliuta coal mine) and the situation with respect to China

Line 92: the minerals mentioned, how have they been identified? In what quantities?

Author Response

Thank you for your valuable suggestions. According to your suggestions, the contents of the paper have been revised and improved, here are response to the changes.

1.Indicate with a map the situation of the samples (Daliuta coal mine) and the situation with respect to China.

The Daliuta Coal Mine in the study area is located on the Bank of the Uranmulun River in Daliuta Town, Shenmu County, Shaanxi Province. Its geographical coordinates are (N39.1 ~N39.4, E111.2 ~E110.5). (This paper is supplemented.) Searching for "Wulan Mulun River" on Google Maps can be found, but the following figure is not listed because of the length of the paper.

2. Line 92: the minerals mentioned, how have they been identified? In what quantities?

The position of each rock stratum is determined by drilling core, and the lithologic parameters are determined by indoor physical and mechanical experiments. This paper supplements the section of main section of 52303 working face in Daliuta Coal Mine, The section of main section of 52303 working face is shown in Fig. 1. The overburden structure of working face is composed of Quaternary loose layer with thickness of 30 m, fine sandstone with thickness of 187 m and siltstone with thickness of 18 m from top to bottom.

Figure 1 The section figure of 52303 work facing

As an effective means of rock engineering research, geomechanical model test has been published by many researchers on the proportion of similar materials. On the basis of previous research results, an experimental model is made according to the physical and mechanical properties of rock strata in Daliuta Coal Mine. The dosage of similar material in related experiments is shown in Table 2.

Reviewer 5 Report

Dear authors,

All my suggestion you can find in attached PDF.

Regards,

Reviewer A

Author Response

First of all, thank you for your valuable suggestions on this paper, so that this article can be improved. Then I will respond to your question.

1. The citation does not ask for names, the number is just enough.

This part has been corrected in the newly uploaded revision.

2. number_space_unit (rule)

All numbers and units are corrected in accordance with the rules.

3. I am pretty sure that you worked with sandstones. Sand can not included consolidated coal strata in normal geological evolution.

Your opinion is very important, I am sorry that the translation was not made clear. This section should be changed to “fine sandstone” and “siltstone”. It has now been modified.

4. “The average inclination of the coal seam in coal face 52303 of Daliuta coal mine is 2°”Geological section or isopach map needs to be given.

This paper supplements the part of main section of 52303 working face in Daliuta Coal Mine, as shown in the following figure.

The dip angle of 2 degrees indicates that the overburden is near horizontal. For the convenience of construction, the test model is laid according to the horizontal strata.

Figure 1 The section figure of 52303 work facing

5. “the geological structure is simple” Which one?

This part is not clearly stated. It should be "the overlying strata of 52303 working face in Daliuta Coal Mine", which has been corrected in this paper.

6.“layer I and II of the third section”The reader still does not know anything about those layers, compositions, space distributions, stage/age, top and bottom... mine plan with pillars locations on Fig. 1 covers too small part of the mine to conclude anything about coal geology settings.

This section was re-described in the revised draft. The description is as follows:

In order to understand the characteristics of overburden deformation caused by coal seam mining, according to the "three zones" discriminant theory, three monitoring layers (I, II, and III) are set up along the strike of the working face at 5 mm, 205 mm and 405 mm away from the top of the coal seam, respectively [18]. According to the disturbance degree caused by coal seam mining, two sensing optical fibers are laid along the center of goaf, the edge of goaf and non-goaf in each monitoring layer. Before the test, the sensing optical fiber at the adjacent position of each layer is fused from the beginning to end. Then, the sensing optical fiber of each layer is connected in series from top to bottom to form one optical fiber.

7. “Figure 4.”Is there connection between this and Fig. 1?

In the model box of Figure 1, layer I and II are laid with six sensing fibers respectively, and the strain of overburden during mining is monitored. The obtained strain is interpolated by Kriging method and the strain contour map is obtained, as shown in Figure 4 and Figure 5.

8. “contour map”Contours are:a) hardly to follow in legend (where units are missing);

b) I do not understand meaning of axis Y (what is tendency)?

c) Which method did you use for interpolation?

d) The maps are full of small "bull-eyes" shapes. Hardly that strain will be so much localised.

(a) Figures 4 and 5 in the original text are strain contour maps in units of 10-6, which have been supplemented by the name of the map.

(b) Y axis is the tendency of working face, that is, the direction of coal seam mining. Through research and analysis, it is found that along the working face tendency, when the reamer-pillar mining area is small, the tension strain and compression strain of overlying strata in the goaf area are all in the center of the mining face, and there is an axisymmetric distribution. With the increase of reamer-pillar mining face, the stress value in the center of goaf is obviously larger than that in the edge of goaf. Some modifications have been made in this paper.

(c) Based on the irregular distribution of data in the monitoring of mining overburden deformation, this paper uses the "Kriging" method with high gridding accuracy to interpolate data.

(d) At present, geophysical prospecting method can analyze the evolution characteristics of the fissures in the goaf, but when the model size is small and the fissures are less developed, the measurement error is large.

In this paper, BOTDA distributed optical fiber sensing technology is used. The spatial resolution of the instrument is 10 cm and the sampling interval is 5 cm. In this paper, six sensing optical fibers are laid in rock layers 1 and 2, and the distance between adjacent parallel optical fibers is 35 cm. The data are processed by difference fitting method. The results show that with the increase of monitoring data, "bull-eyes" phenomenon is obvious. The main reason is that the measurement accuracy of BOTDA instrument is (+25 u epsilon), and its accuracy is high. The phenomenon of "bull-eyes" is related to the setting of data between contours, that is, with the increase of strain, the denser the contours are. Secondly, the laying spacing of sensing optical fibers in overlying strata is related. If the distance between sensing fibers is reduced, the phenomenon of "bull-eyes" can be reduced.

9.“the maximum compressive strain is concentrated”Each statement support by values from map.

It has been explained in this paper.

Round 2

Reviewer 1 Report

Dear Authors, Thanks for your efforts for revision. It seems the paper has the better shape!

Author Response

thank you for your valuable suggestions on this paper.

Reviewer 3 Report

It is ready to publish.

Author Response

(The authors gave the same response as above.)

Reviewer 5 Report

Dear author,

Each Kriged set of maps needs to be shown with variorum model(s). The "bull-eyes" effect can be significantly reduced with well prepared variograms, whatever density of measurements had been used.

Regards

Author Response

The strain contour diagram in this paper represents the change process of overburden strain caused by coal pillar mining. I will seriously consider your suggestions and thank you again for your good suggestions for this article.